# A variant-centric perspective on geographic patterns of human allele frequency variation

Arjun Biddanda, Daniel P Rice, John Novembre*

Department of Human Genetics, University of Chicago, Chicago, United States

**Abstract** A key challenge in human genetics is to understand the geographic distribution of human genetic variation. Often genetic variation is described by showing relationships among populations or individuals, drawing inferences over many variants. Here, we introduce an alternative representation of genetic variation that reveals the relative abundance of different allele frequency patterns. This approach allows viewers to easily see several features of human genetic structure: (1) most variants are rare and geographically localized, (2) variants that are common in a single geographic region are more likely to be shared across the globe than to be private to that region, and (3) where two individuals differ, it is most often due to variants that are found globally, regardless of whether the individuals are from the same region or different regions. Our variant-centric visualization clarifies the geographic patterns of human variation and can help address misconceptions about genetic differentiation among populations.

## Introduction

Understanding human genetic variation, including its origins and its consequences, is one of the long-standing challenges of human biology. A first step is to learn the fundamental aspects of how human genomes vary within and between populations. For example, how often do variants have an allele at high frequency in one narrow region of the world that is absent everywhere else? For answering many applied questions, we need to know how many variants show any particular geographic pattern in their allele frequencies.

In order to answer such questions, one needs to measure the frequencies of many alleles around the world without the ascertainment biases that affect genotyping arrays and other probe-based technologies (*International HapMap Consortium, 2005*; *Li et al., 2008*). Recent whole-genome sequencing studies (*Auton et al., 2015*; *Mallick et al., 2016*; *Bergström et al., 2019*; *Fairley et al., 2020*) provide these data, and thus present an opportunity for new perspectives on human variation.

However, large genetic data sets present a visualization challenge: how does one show the allele frequency patterns of millions of variants? Plotting a joint site frequency spectrum (SFS) is one approach that efficiently summarizes allele frequencies and can be carried out for data from two or three populations (*Gutenkunst et al., 2009*). For more than three populations, one must resort to showing multiple combinations of two or three-population SFSs. This representation becomes unwieldy to interpret for more than three populations and cannot represent information about the joint distribution of allele frequencies across all populations. Thus, we need visualizations that intuitively summarize allele frequency variation across several populations.

New visualization techniques also have the potential to improve population genetics education and research. Many commonly used analysis methods, such as principal components analysis (PCA) or admixture analysis, do a poor job of conveying absolute levels of differentiation (*McVean, 2009*; *Lawson et al., 2018*). Observing the genetic clustering of individuals into groups can give a misleading impression of 'deep' differentiation between populations, even when the signal comes from

*For correspondence:
jnovembre@uchicago.edu

Competing interests: The authors declare that no competing interests exist.

subtle allele frequency deviations at a large number of loci (*Patterson et al., 2006*; *McVean, 2009*; *Novembre and Peter, 2016*). Related misconceptions can arise from observing how direct-to-consumer genetic ancestry tests apportion ancestry to broad continental regions. One may mistakenly surmise from the output of these methods that most human alleles must be sharply divided among regional groups, such that each allele is common in one continental region and absent in all others. Similarly, one might mistakenly conclude that two humans from different regions of the world differ mainly due to alleles that are restricted to each region. Such misconceptions can impact researchers and the broader public alike. All these misconceptions potentially can be avoided with visualizations of population genetic data that make typical allele frequency patterns more transparent.

Here, we develop a new representation of population genetic data and apply it to the New York Genome Center deep coverage sequencing data of the 1000 Genomes Project (1KGP) samples (*Auton et al., 2015*). In essence, our approach represents a multi-population joint SFS with coarsely binned allele frequencies. It trades precision in frequency for the ability to show several populations on the same plot. Overall, we aimed to create a visualization that is easily understandable and useful for pedagogy. As we will show, the visualizations reveal with relative ease many known important features of human genetic variation and evolutionary history.

This work follows in the spirit of *Rosenberg, 2011* who used an earlier dataset of microsatellite variation to create an approachable demonstration of major features in the geographic distribution of human genetic variation (as well as earlier related papers such as *Lewontin, 1972*; *Mountain and Ramakrishnan, 2005*; *Witherspoon et al., 2007*). Our results complement several recent analyses of single-nucleotide variants (SNVs) in whole-genome sequencing data from humans (*Auton et al., 2015*; *Mallick et al., 2016*; *Bergström et al., 2019*). We label the approach taken here a variant-centric view of human genetic variation, in contrast to representations that focus on individuals or populations and their relative levels of similarity.

## Materials and methods

To introduce the approach, we begin with considering 100 randomly chosen SNVs sampled from Chromosome 22 of the 1KGP high coverage data (*Box 1*, *Fairley et al., 2020*). *Figure 1* shows the allele frequency of each variant (rows) in each of the 26 populations of the 1KGP (columns, see *Supplementary file 1* for labels). As a convention throughout this paper, we use darker shades of blue to represent higher allele frequency, and we keep track of the globally minor allele, that is, the rarer (<50% frequency) allele within the full sample. The figure shows that variants seem to fall into a few major descriptive categories: variants with alleles that are localized to single populations and rare within them, and variants with alleles that are found across all 26 populations and are common within them.

To investigate whether such patterns hold genome-wide, we devise a scheme that allows us to represent the >90 million SNVs in the genome-wide data (*Figure 2*). First, we follow the 1KGP study in grouping the samples from the 26 populations into five geographical ancestry groups: African (AFR), European (EUR), South Asian (SAS), East Asian (EAS), and Admixed American (AMR) (*Figure 2A*, *Box 1*). For clarity,

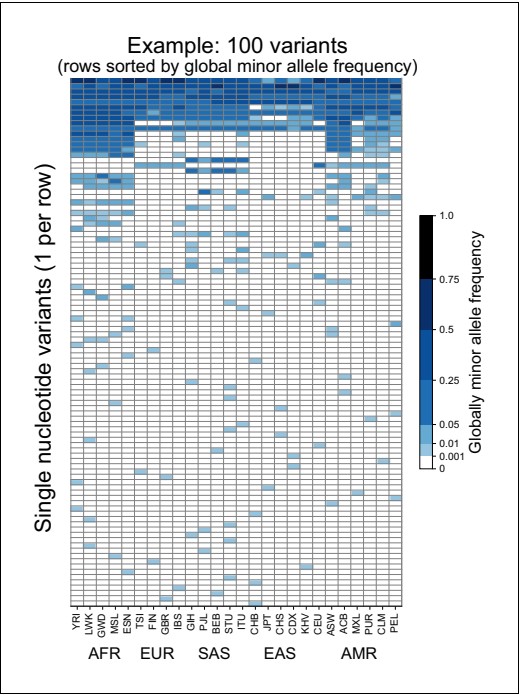

**Figure 1.** Allele frequencies at 100 randomly chosen variants from Chromosome 22. Frequencies of the globally minor allele are shown across 26 populations (columns) from the 1KGP for 100 randomly chosen variants from Chromosome 22. Note that the allele frequency bin spacing is nonlinear to capture variation at low as well as high frequencies. Populations are ordered by broad geographic region (horizontal labels, see *Figure 2A* for legend). Definitions of abbreviations for the 26 1KGP populations are given in *Supplementary file 1*.

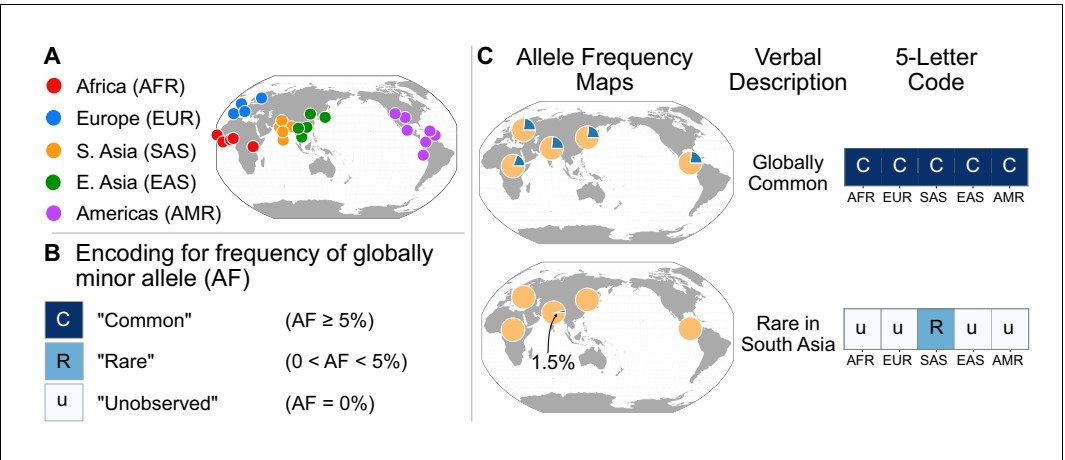

**Figure 2.** A simple coding system to represent geographic distributions of variants. (**A**) Regional groupings of the 26 populations in the 1KGP Project. (**B**) Legend for minor allele frequency bins. (**C**) Two examples of how a verbal description of an allele frequency map can be communicated equivalently with a five-letter code (yellow signifies the major allele frequency, blue signifies the minor allele frequency in the pie charts).

The online version of this article includes the following figure supplement(s) for figure 2:

**Figure supplement 1.** Probability of not observing a variant at a given allele frequency and sample size in number of individuals.

we modify the original 1KGP groupings slightly for this project (by including several samples from the Americas in the AMR grouping, see *Box 1*). While human population structure can be dissected at much finer scales than these groups (e.g. *Leslie et al., 2015*; *Novembre and Peter, 2016*), the regional groupings we use are a practical and instructive starting point—as we will show, several key

## Box 1. Dataset descriptions and groupings.

We use bi-allelic single-nucleotide variants from the New York Genome Center high-coverage sequencing of the 1000 Genomes Project (1KGP) Phase 3 samples (*Auton et al., 2015*) (see key resources table, accessed July 22nd, 2019, only variants with PASS in the VCF variant filter column). Most of the samples are from an ethnic group in an area (e.g. the 'Yoruba of Ibadan,' YRI, or the 'Han Chinese from Beijing,' CHB), so the sampling necessarily represents a simplification of the diversity present in any locale (e.g. Beijing is home to several ethnic groups beyond the Han Chinese). For each grouping, the 1KGP typically required each individual to have at least three of four grandparents who identified themselves as members of the group being sampled.

The 1KGP further defined five geographical ancestry groups: African (AFR), European (EUR), South Asian (SAS), East Asian (EAS), and Admixed American (AMR). Differing from the 1KGP, we include in the 'Admixed in the Americas' (AMR) regional grouping the following populations: 'Americans of African Ancestry in SW USA', 'African-Caribbeans in Barbados (ACB)', and the 'Utah Residents (CEPH) with Northern and Western European Ancestry'. We chose this grouping because it is a more straightforward representation of current human geography. See *Supplementary file 1* for a full list of the 26 populations and the grouping into five regions. We note challenges and caveats of these alternate decisions in the Discussion. Also, *Figure 6* and *Figure 5—figure supplements 1–3* provide a complementary view to *Figure 3B, C* and *Figures 4* and *6*, where the analysis is not based on the five groupings, but instead all 26 populations.

features of human evolutionary history become apparent, and many misconceptions about human differentiation can be addressed efficiently with this coarse approach (see Discussion). As any such groupings are necessarily arbitrary, we also show results without using regional groupings to calculate frequencies (see section 'Finer-scale resolution of variant distributions' below).

To represent the geographic distributions of alleles compactly, we give every variant a five-letter code according to its allele frequencies across regions (*Figure 2A*). More precisely, for each bi-allelic SNV, we identify the global rarer (minor) allele. Then for each region, we code the allele's frequency as 'u', 'R', or 'C', based on whether the allele is '(u)ndetected,' '(R)are,' or '(C)ommon' (*Figure 2B*). To distinguish between 'rare' and 'common' alleles, we used a threshold of 5% frequency. Finally, we concatenate the allele's regional frequency codes in the fixed (and arbitrary) order: AFR, EUR, SAS, EAS, and AMR. This procedure generates a 'geographic distribution code' for each variant. For example, the code 'CCCCC' represents a variant that is common across every region, while 'uuRuu' represents a variant that is rare in South Asia and unobserved elsewhere (*Figure 2C*). To display the relative abundance of codes within a set of variants, we use a vertical stack from the most abundant code at the bottom to the least abundant at the top, with the height of each code proportional to its abundance, so that the cumulative proportions of the rank-ordered codes are easily readable (*Figure 3*).

## Results

Using the encoding scheme just described, we generated geographic distribution codes for all ~92 million biallelic SNVs in the 1000 Genomes dataset and display their relative proportions (*Figure 3*). The distribution of codes is heavily concentrated, with 85% of variants falling into just eight codes out of the 242 that are possible ($3^5$–1: three frequency categories in each of five regional groupings, subtracting the code 'UUUUU' as each variant has been observed by definition). Of the top eight codes, the top four codes represent rare variants that are localized in a single region. The fifth most abundant code, 'RuuuR', represents rare variants found in Africa and the Admixed Americas (which includes African American individuals, for example). The sixth code is another set of localized rare variants ('uRuuu', i.e. variants rare in EUR). The seventh code is 'CCCCC' or 'globally common variants.' The eighth most abundant category, 'uRuuR', represents rare variants found in Europe and the Admixed Americas. Conspicuously infrequent in the distribution are variants that are common in only one region outside of Africa and absent in others (e.g. 'uCuuu', 'uuCuu', 'uuuCu', 'uuuuC'). Instead, when a variant is found to be common (>5% allele frequency) in one population, the modal pattern (37.3%) is that it is common across the five regions ('CCCCC'). Further, 63% of variants common in at least one region are also globally widespread, in the sense of being found across all five regions. This number rises to 82% for variants common in at least one region outside of Africa (*Figure 3—figure supplements 1* and *2*).

Singleton variants—alleles found in a single individual—are the most abundant type of variant in human genetic data and are necessarily found in just one geographic region. To focus on the distributions of non-singleton variants, we removed singletons and tallied again the relative abundance of patterns (*Figure 3C*). Removing singletons reduces the absolute number of variants observed by 48.2% (91,784,637 vs. 44,290,364). Without singletons, we see more clearly the abundance of patterns that have rare variants shared between two or more regions (codes with two 'R's and one 'u', such as 'uuRRu' or 'RRuuu').

The scheme for geographic distribution codes requires a few choices. For comparison, we show results using a 1% minor allele frequency threshold to define 'common' variants (*Figure 3—figure supplement 3A*). We also produced results tracking the derived (younger) rather than the globally minor allele (*Figure 3—figure supplement 3C*; for 96.6% of variants in the dataset with high-quality ancestral allele calls [*Box 1*], the globally minor allele is the derived allele). Neither changing the frequency threshold to 1% nor tracking the derived allele meaningfully affects the major patterns observed.

The patterns observed here are interpretable in light of some basic principles of population genetics. Rare variants are typically the result of recent mutations (*Mathieson and McVean, 2014*; *Kiezun et al., 2013*; *Kimura and Ohta, 1973*; *Albers and McVean, 2020*). Thus, we interpret the localized rare variants (such as 'Ruuuu' or 'uuuRu') as mostly young mutations that have not had time to spread geographically. The code 'CCCCC' (globally common variants), likely comprises mostly older variants that arose in Africa and were spread globally during the Out-of-Africa migration and other dispersal events (see *Box 2*). The appearance of rare variants shared between two or more

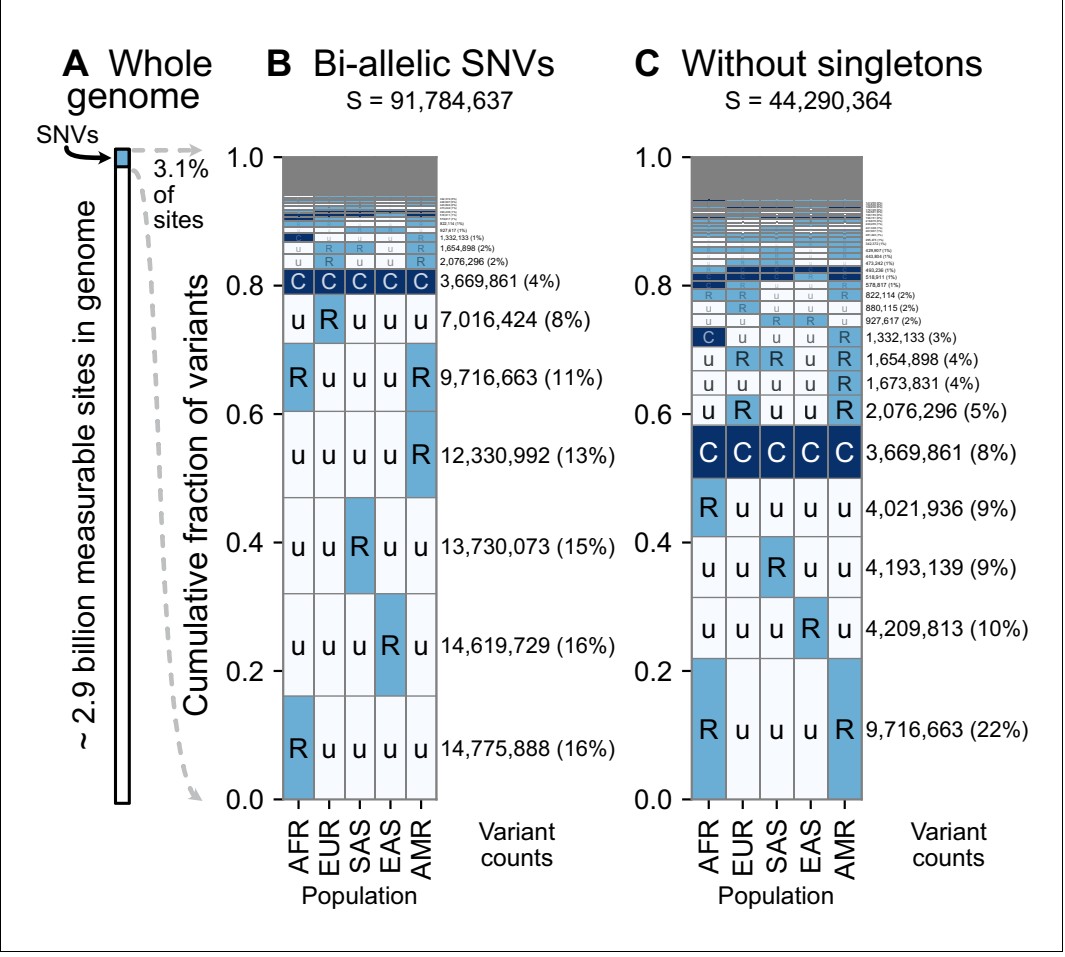

**Figure 3.** A summary of geographic distributions in human SNVs. (**A**) We observe variants at ~3.1% of the measurable sites in the reference human genome (GRCh38). A measurable site is one at which it is possible to detect variation with current sequencing technologies (currently approximately 2.9 Gb out of 3.1 Gb in the human genome; ). (**B and C**) The relative abundance of different geographic distributions for 1KGP variants, (**B**) including singletons, and (**C**) excluding singletons. In panels B and C, the right-hand rectangles show the number and percentage of variants that fall within the corresponding geographic code on the left-hand side; distribution patterns are sorted by their abundance, from bottom-to-top. See *Figure 2* for an explanation of the five-letter 'u', 'R', 'C' codes. The proportion of the genome with variants that have a given geographic distribution code can be calculated from the data above (for example, with the 'Ruuuu' code, as 17% × 3.1% = 0.53%). The gray box represents geographic distribution codes whose abundances are too rare to effectively display at the given figure resolution.

The online version of this article includes the following figure supplement(s) for figure 3:

**Figure supplement 1.** Alternate versions of the GeoVar plots with an alternateallele frequency threshold and tracking derived versus minor allele frequencies.

**Figure supplement 2.** Proportion of variants with specific GeoVar patterns conditional on an allele being common in at least one continental group.

**Figure supplement 3.** Proportion of variants with specific GeoVar patterns conditional on an allele being 'globally widespread'.

**Figure supplement 4.** GeoVar plots derived from simulations of two published models of human demography.

regions (codes with two 'R's and three 'u's, such as 'uuRRu' or 'RRuuu') is likely the signature of recent gene flow between those regions (*Box 2*; *Platt et al., 2019*; *Mathieson and McVean, 2014*; *Gutenkunst et al., 2009*). In particular, the abundant 'RuuuR' and 'uRuuR' codes likely represent young variants that are shared between the Admixed Americas and Africa ('RuuuR') or Europe ('uRuuR') because of the population movements during the last 500 years that began with European

colonization of the Americas and the subsequent slave trade from Africa. We interpret the 10th most abundant code ('CuuuR', *Figure 3B*) as mostly variants that were lost in the Out-of-Africa bottleneck and subsequently carried to the Americas by African ancestors. There is a relative absence of variants that are common in only one region outside of Africa and absent across all others (e.g. 'uCuuu', 'uuCuu', 'uuuCu', 'uuuuC'). These patterns are consistent with human populations having not diverged deeply, in the sense that there has not been sufficient time for genetic drift to greatly shift allele frequencies among them (*Box 2*). To help make this clear, consider the alternative scenario—a model with very ancient population splits (*Coon, 1962*). In such a model, one would expect many more variants to be common to one region and absent in others ('Cuuuu' or 'uuuCu' for example, see *Box 2*). Overall, these results reflect a timescale of divergence consistent with the Recent-African-Origin model of human evolution as well as subsequent gene flow among regions (*Cann et al., 1987*; *Stringer and Andrews, 1988*; *Thomson et al., 2000*; *Ramachandran et al., 2005*; *Pickrell and Reich, 2014*).

## The variants that differ between a pair of individuals

While *Figure 3* illustrates genetic variants found in a large, global sampling of human diversity, it does not show what to expect for the variants that differ between pairs of individuals. Are the variants that differ between two individuals more often geographically widespread or spatially localized?

To address this question, we considered the variants carried by pairs of individuals from the whole-genome sequencing data of the Simons Genome Diversity Project (SGDP) (*Mallick et al., 2016*; *Figure 4*). The SGDP sampled 300 individuals from 142 diverse populations. We use the SGDP data to avoid ascertainment biases that might arise from looking at individuals within the same dataset we use to measure allele frequencies. *Figure 4* shows a representative subset with six

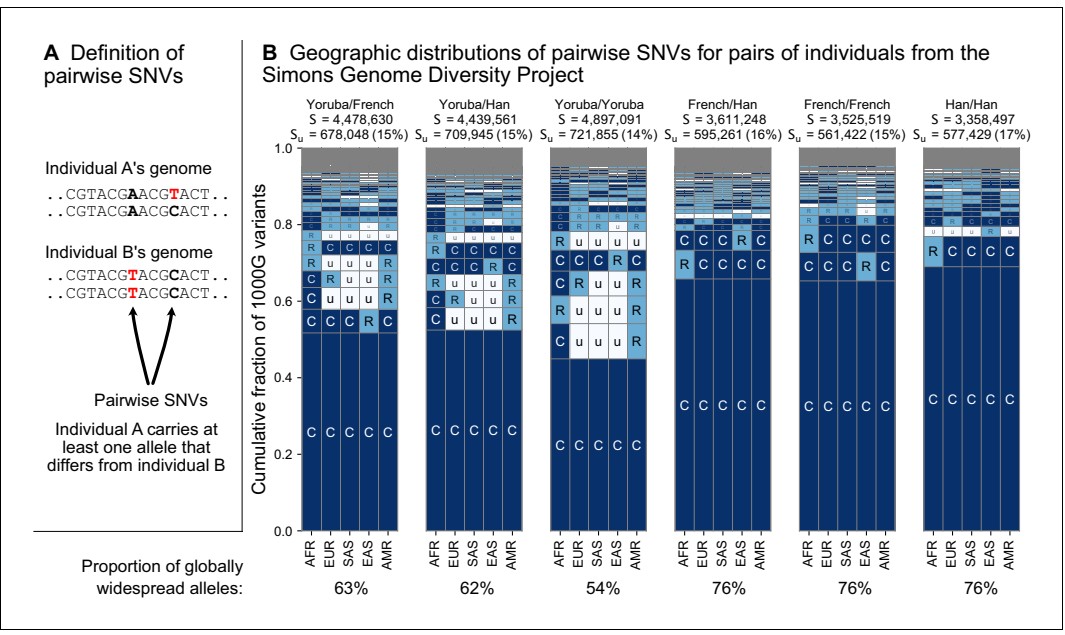

**Figure 4.** The geographic distributions of SNVs between pairs of individuals. (**A**) Definition of a pairwise SNV. (**B**) The abundance of geographic distribution codes for different pairs of individuals from the SGDP dataset. Above each plot, we show the total number of variants that differ between each individual ($S$) and the number that were unobserved completely in the 1KGP data ($S_U$). Across the bottom, we show the proportion of variants with globally widespread alleles for each pair. We calculate this as the fraction of variants with no 'u' encodings over the total number of variants ($S$). (Note: by doing so, we make the assumption that if a variant is not found in the 1KGP data it is not globally widespread). For this analysis, as in *Mallick et al., 2016*, we include only autosomal biallelic SNVs for variants that pass 'filter level 1'.

The online version of this article includes the following figure supplement(s) for figure 4:

**Figure supplement 1.** Additional examples of geographic distribution codes for pairwise variants from different pairs of sampled individuals in the SGDP.

pairs chosen from three populations (*Figure 4—figure supplement 1*, shows a larger set of examples). For each pair, we see some variants that were undiscovered in the 1KGP data (denoted $S_u$ in the figure). These account for 17–20% of each set of pairwise SNVs and are likely rare variants. We see that the variants that differ between each pair of individuals are typically globally widespread (i.e. codes with no 'u's, with proportions out of the total S varying from 54% to 76% for the pairs in *Figure 4*). The observation of mostly globally common variants in pairwise comparisons may seem counterintuitive considering the abundance of rare, localized variants overall. However, precisely because rare variants are rare, they are not often carried by either individual in a pair. Instead, pairs of individuals mostly differ because one of them carries a common variant that the other does not; and as *Figure 3* already showed, common variants in any single location are often common throughout the world (also see Figure 6 and *Figure 3—figure supplement 3*).

From the example pairwise comparisons (*Figure 4*, and *Figure 4—figure supplement 1*), one also observes evidence for higher diversity in Africa, which is typically interpreted in terms of founder effects reducing diversity outside of Africa (*Cann et al., 1987*; *Harpending and Eller, 2000*; *Harpending and Rogers, 2000*; *Ramachandran et al., 2005*; *Prugnolle et al., 2005*), although other models, especially ones including substantial subsequent admixture, can also produce this pattern (*DeGiorgio et al., 2009*; *Pickrell and Reich, 2014*). For example, the two Yoruba individuals have more pairwise SNVs (S = 4,897,091) than the French/French (S = 3,525,519) and Han/Han (S = 3,358,497) pairs. Pairs involving one or both of the sample Yoruba individuals have more variants with alleles common in Africa and rare or absent elsewhere (e.g. 'CuuuR',' RuuuR'). Finally, a more subtle, but expected, impact of founder effects is that the sample Yoruba/Yoruba comparison is expected to have higher numbers of pairwise variants than the sample Yoruba/Han or Yoruba/French comparison, which we observe.

## The geographic distributions of variants typed on genotyping arrays

Targeted genotyping arrays are a cost-effective alternative to whole-genome sequencing. In contrast to whole-genome sequencing, genotyping arrays use targeted probes to measure an individual's genotype only at preselected variant sites. The process of discovering and selecting these target sites typically enriches the probe sets toward common variants (*Clark et al., 2005*), underrepresents geographically localized variants (*Albrechtsen et al., 2010*; *Lachance and Tishkoff, 2013*), and can affect genotype imputation and genetic risk prediction (*Howie et al., 2012*; *Martin et al., 2017*).

*Figure 5* shows the geographic distributions of bi-allelic SNVs included on five popular array products. In stark contrast with the SNVs identified by whole-genome sequencing (*Figure 3B*), a large fraction of the variants on genotyping arrays are globally common. This is especially true for the Affy6, Human Origins, and OmniExpress arrays, which were designed using polymorphisms ascertained from a smaller number of sequenced individuals, and primarily capture more common variants due to this ascertainment. The Omni2.5Exome and MEGA arrays in contrast exhibit many more rare variants. In both these arrays, the second and third most abundant codes are 'CuuuR' and 'RuuuR' variants. The MEGA array was uniquely designed to capture rare variation in undersampled continental groups, including African ancestries (*Bien et al., 2016*; *Bien et al., 2019*). *Wojcik et al., 2019* found that this design improved African and African American imputation accuracy, leading to greater power to map population-specific disease risk.

## Finer-scale resolution of variant distributions

While the use of five regional groupings above allows us to describe variant distributions compactly with a five-digit encoding, the basic principle of grouping allele frequencies can be extended to build a 26-digit encoding for the 1KGP variants (*Figure 6*, *Figure 6—figure supplements 1–3*). Doing so with the set of ~92 million variants found in the 1KGP project (*Figure 6*), we find a consistent pattern with *Figure 3B*, in that the majority of variants are seen to be rare and geographically localized (1 'R', and the remainder 'u's), and when a variant is common in any one population, it is typically common across the full set of populations (*Figure 6*, pattern with all 'C's). This view reveals that the five-digit encodings with 1 'R' and 4 'u's are often due to variants that are rare even within a single population. This is not unexpected given many of them are singletons. When we remove singletons (*Figure 6—figure supplement 1B*), we again see more clearly rare allele sharing indicative of recent gene flow, although at finer-scale resolution.

## Box 2. Theoretical modeling.

We can use theoretical models to estimate what our visualizations would look like for two populations in simple contrasting cases of 'deep' divergence, 'shallow' divergence, and 'shallow' divergence with gene flow. The shallow case is calibrated to be qualitatively consistent with the Recent-African-Origin model with subsequent gene flow. The deep case mimics inaccurate models of human evolution with very ancient population splits (e.g. **Coon, 1962**). For each case, we computed the expected abundances of distribution codes in a simple model of population divergence: two modern populations of $N$ individuals each that diverged $T$ generations ago from a common population of $N$ individuals (see Appendix 1 for information about this calculation). We model gene flow by including recent admixture: individuals in Population A derive an average fraction $\alpha$ of their ancestry from Population B and vice versa. This simplified model neglects many of the complications of human population history, including population growth, continuous historical migration, and natural selection, but it captures the key features of common origins, divergence, and subsequent contact (see **Figure 3—figure supplement 4** to compare with simulation results from more complex published models of human population history).

In this model, the key control parameter is $T/2N$, the population-scaled divergence time. Human pairwise nucleotide diversity ($\sim 1 \times 10^{-3}$) and per-base-pair per-generation mutation rate ($\sim 1.25 \times 10^{-8}$) imply a Wright-Fisher effective population size of $N = 2 \times 10^4$ individuals. The Out-of-Africa divergence is estimated to have occurred approximately 60,000 years ago (**Nielsen et al., 2017**). Assuming a 30-year generation time (**Fenner, 2005**) gives $T/2N = 0.05$. We compare this scenario with $T/2N = 0.5$, corresponding to a deeper divergence of approximately 600,000 years ago.

**Box 2—figure 1A** shows the expected patterns in a sample of 100 individuals from each population for deep divergence ($T/2N = 0.5$), shallow divergence ($T/2N = 0.05$) without admixture, and shallow divergence with admixture ($\alpha = 0.02$). The shallow divergence model with or without admixture reproduces the preponderance of 'Ru' and 'CC' mutations seen in the data, while the deep divergence model shows many more 'Cu' and many fewer 'CC' mutations. The case with admixture shows a slight increase in variant sharing ('RR' alleles increase from 1.3% of variants to 4.2%; 'RC' and 'CR' alleles increase from 6% to 10%; 'CC' alleles comprise 23% in both cases).

We can understand the relationship between the split time and geographic distribution abundances heuristically as follows. During an interval of $\Delta t$ generations, the frequency of a neutral mutation starting at frequency $f$ changes randomly by a typical amount $\Delta f \sim \sqrt{\frac{f(1-f)}{2N}} \Delta t$. Consider a mutation that is at 25% frequency, that is, common, in the ancestral population at the time of the split (**Box 2—figure 1B**). At time $\Delta t/2N = 0.05$ after the split, the frequency of the mutation is likely to be in the interval (15%, 35%) in both populations and will be assigned the code 'CC'. On the other hand, by time $\Delta t/2N = 0.5$ after the split, the mutation has a significant chance of going extinct in one or both populations (**Box 2—figure 1C**). Mutations that go extinct in one population but not the other will typically be assigned a code 'Cu' or 'uC'. At the same time, new mutations are constantly entering the evolving populations. These new mutations will be private to one population ('Ru' or 'Cu') and the overwhelming majority will go extinct before reaching detectable frequencies. Conditional on non-extinction, the expected frequency of a neutral mutation increases linearly with time (see Appendix 2). As a result, the frequencies of new mutations since the split time $\Delta t$ will mostly be contained in a triangular envelope $f < \Delta t/2N$ (**Box 2—figure 1B**). For recent divergence, the new mutations will be assigned code 'Ru' or 'uR', while in deeply diverged populations they may be categorized as 'Cu' or 'uC'.

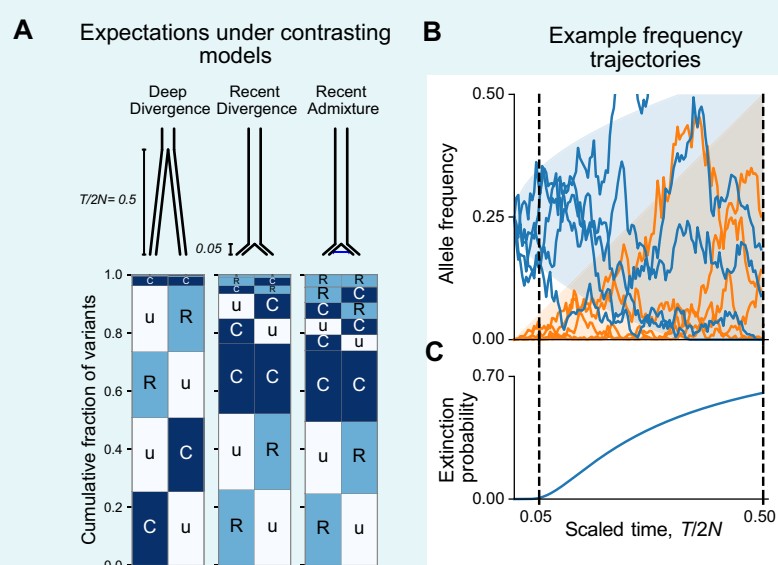

**Box 2—figure 1.** Allele frequency patterns depend on the time since population divergence and levels of admixture.
(**A**) Expected geographic distribution code abundances in a sample of 100 diploid individuals from each of two populations, for deep divergence ($T/2N = 0.5$, $\alpha = 0$), recent divergence without admixture ($T/2N = 0.05$, $\alpha = 0$), and recent divergence with admixture ($T/2N = 0.05$, $\alpha = 0.02$). (**B**) Simulated allele frequency time series for mutations starting at 25% frequency (blue) and new mutations entering the population since the split (orange). (**C**) The probability of extinction of a mutation starting at 25% frequency (see Appendix 2).

## Discussion

By encoding the geographic distributions of the ~92 million biallelic SNVs in the 1KGP data and tallying their abundances, we have provided a new visualization of human genetic diversity. We term our figures 'GeoVar' plots as they help reveal the geographic distribution of sets of variants. GeoVar plots can complement other methods of visualizing population structure, including: plots of pairwise genetic distance, dimensionality-reduction approaches such as PCA, admixture proportion estimates such as STRUCTURE, and explicitly spatial methods that use the sampling locations of individuals (*Guillot et al., 2009*; *Novembre and Peter, 2016*; *Bradburd and Ralph, 2019*). These previously developed methods help reveal population structure, infer genetic ancestry, and measure historical migration patterns. However, they do a poor job of showing how alleles are distributed geographically. To minimize confusion about levels of differentiation among populations, researchers and educators can consider complementing PCA or STRUCTURE-like outputs with a variant-centric visualization like the ones presented here. To that end, we provide source code to replicate our figures and to generate similar plots for other datasets (the 'GeoVar' software package; see key resources table).

A goal of our work was to build a visualization that can help correct common misconceptions about human genetic variation. First, because many existing methods to describe population structure emphasize between-group or between-individual differentiation, they can convey a misleading impression of 'deep' divergence between populations when it may not exist. Comparing *Figure 1* to outputs of models with 'deep' or 'shallow' divergence can help teach how patterns of human variation are consistent with shallow divergence and the Recent African Origins model (*Box 2*). Second, because personal ancestry tests can identify ancestry to broad continental regions, it is possible to

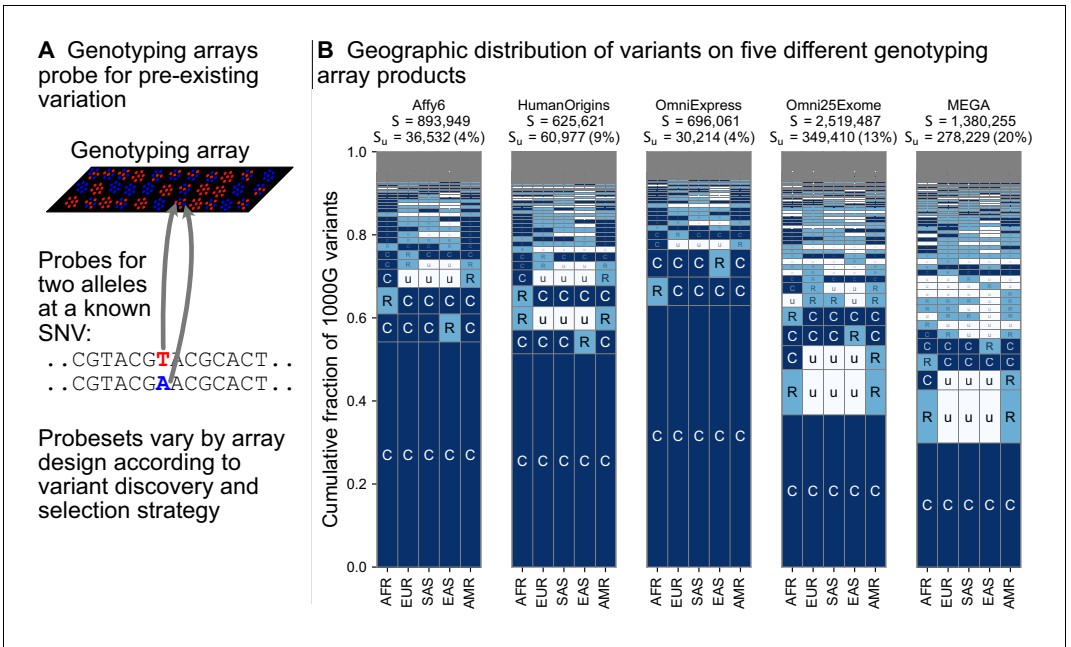

**Figure 5.** Geographic distribution for variants found on genotyping array products. (A) Genotyping arrays consist of probes for a fixed set of variants chosen during the design of the array product. (B) For each array product, we extracted the genomic position of variants found on the array and kept variants that are also found within the 1KGP to highlight their geographic distributions. The arrays considered are the Affymetrix 6.0 (Affy6) genotyping array, the Affymetrix Human Origins array (HumanOrigins), the Illumina HumanOmniExpress (OmniExpress) array, the Illumina Omni2.5Exome, and the Illumina MEGA array. This plot is analogous to Figure 3B but rather than calculating frequencies with the five regional groupings, we compute them within each of the 26 1KGP populations. The total number of variants represented is the same as in *Figure 3B* (S = 91,784,367). See *Figure 2* for an explanation of the 'u','R','C' codes.

incorrectly conclude human alleles are typically found exclusively in a single region and at high frequency within that region (e.g. patterns such as 'uuCuu'.) As our figures show, this is not the case. It should be kept in mind that most fine-scale personal ancestry tests use genotyping arrays and combine evidence from subtle fluctuations in the allele frequencies of many common variants (*Novembre and Peter, 2016*). Finally, another related misconception is that two humans from different regions of the world differ mainly due to alleles that are typical of each region. As we show in *Figure 4*, most of the variants that differ between two individuals are variants with alleles that are globally widespread. (Our awareness of these misconceptions comes from personal experiences in teaching and outreach. However, there is a growing body of formal research on misconceptions regarding human genetic variation, e.g., *Bowling et al., 2008*; *Phelan et al., 2014*; *Hubbard, 2017*; *Roth et al., 2020*).

Our method requires computing allele frequencies within predefined groupings. Grouping and labeling strategies vary between genetic studies and are determined by the goals and constraints of a particular study (*Race, Ethnicity, and Genetics Working Group, 2005*; *Panofsky and Bliss, 2017*; *Mathieson and Scally, 2020*). While we chose deliberately coarse grouping schemes to address the misconceptions described above, the key facts we derive about human genetic variation are robust and appear in finer-grained 26-population versions of the plot (*Figure 6*). We recommend that any application of the GeoVar approach needs to be interpreted with the choice of groupings in mind.

The visualization method developed here is also useful for comparing the geographic distributions of different subsets of variants, (e.g. *Figure 4*, *Figure 5*). For example, when applied to the list of variants targeted by a genotyping array (*Figure 5*), the approach quickly reveals the relative balance of common versus rare variants and the geographical patterns of those variants.

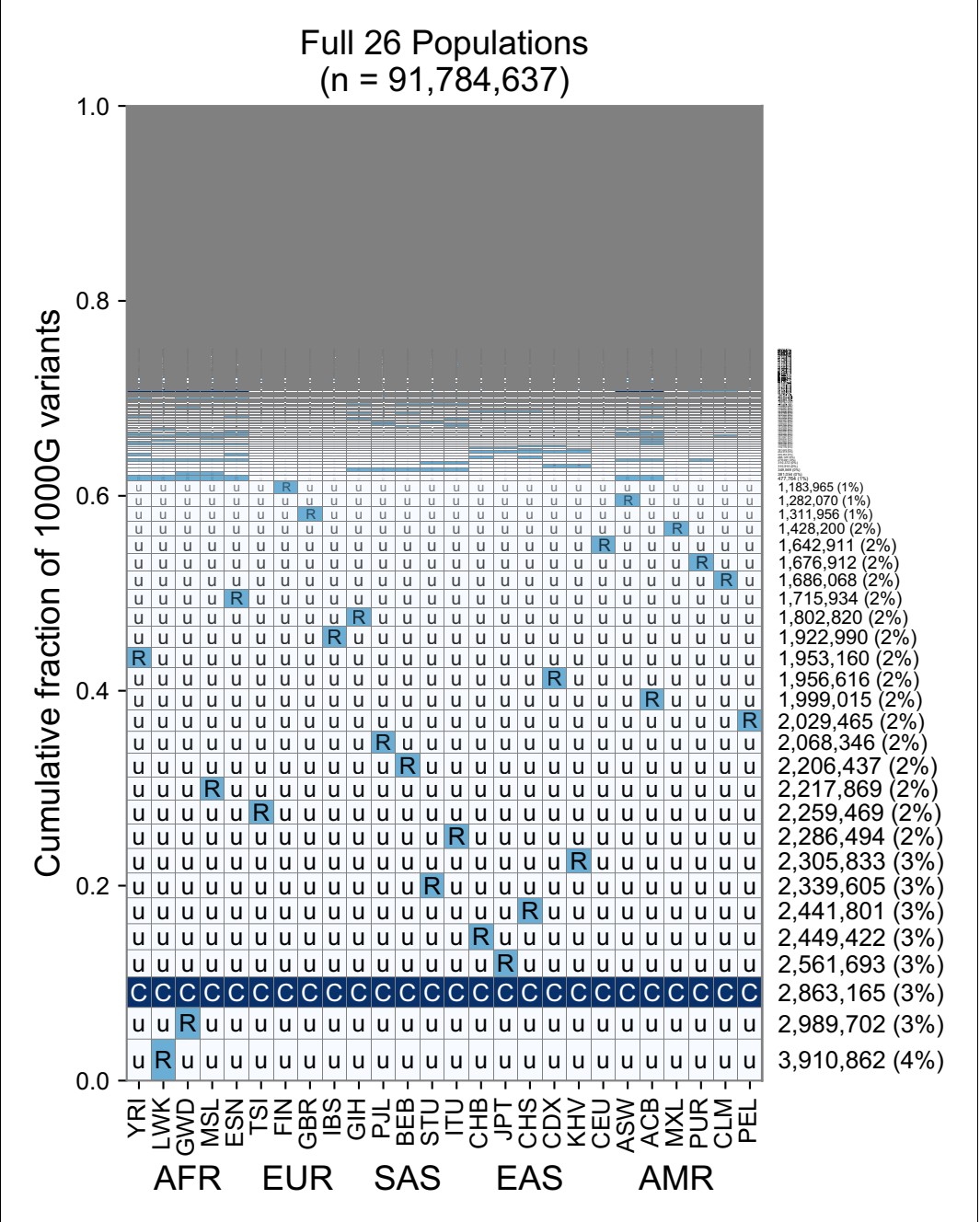

**Figure 6.** A finer-scale summary of geographic distributions in human SNVs from the 1KGP. This plot is analogous to *Figure 3B* but rather than calculating frequencies with the five regional groupings, we compute them within each of the 26 1KGP populations. The total number of variants represented is the same as in *Figure 3B* (S = 91,784,367). See *Figure 2* for an explanation of the 'u','R','C' codes.

The online version of this article includes the following figure supplement(s) for figure 6:

**Figure supplement 1.** The geographic distribution of variants across all 26 populations
(for legend see *Supplementary file 1*) in the 1KGP both with singletons included (**A**) and removed (**B**).

**Figure supplement 2.** The geographic distribution of pairwise SNVs across pairs of individuals from the Simons Genome Diversity Project using the full set of 26 populations from the 1KGP.

**Figure supplement 3.** The geographic distribution of SNVs on genotyping s using the full set of 26 populations from the 1KGP.

**Figure supplement 4.** The minor allele frequencies of 300 variants in each of the 26 original population labels in the 1KGP.

Interpreting the results of this visualization approach does have some caveats. First, we estimate the frequency of alleles from samples of local populations. We expect that as sample sizes increase many alleles called as unobserved 'u' will be reclassified as rare 'R'. The average sample size across all of our geographic regions is approximately 500 individuals (AFR: 504, EUR: 404, SAS: 489, EAS: 504, AMR: 603). Assuming regions are internally well-mixed, we have ~80% power to detect alleles with a frequency of ~0.2% in a region (*Figure 2—figure supplement 1*). For alleles with lower frequencies, we would require larger sample sizes to ensure similar detection power (*Figure 2—figure supplement 1*). An implication is that in large samples, we should observe more rare variant sharing. Thus, we expect the figures here to underrepresent the levels of rare variant sharing between human populations. In general, one must keep in mind that the GeoVar plot is a visualization of the joint SFS for the sample, rather than for the complete population.

A second caveat is that our encoding groups a wide range of variants into the '(C)ommon' category (i.e. all variants where the frequency of the globally minor allele is greater than 5%). For some applications, such as population screening for carriers, it may be enough to know that a variant falls in the 'rare' or 'common' bins we have described, and more detail is inconsequential. For other applications, the detailed fluctuations in allele frequency across populations are relevant—for example, differences in allele frequencies at common variants (*Figure 6—figure supplement 4*) are regularly used to infer patterns of population structure and relatedness (*Li et al., 2008*; *Pickrell and Pritchard, 2012*; *Patterson et al., 2012*).

Third, one must interpret our results with the sampling design of the 1KGP study design in mind. In particular, the 1KGP filtered for individuals of a single ethnicity within each locale. However, in our current cosmopolitan world, the genetic diversity in any location or broad-based sampling project will be considerably higher than implied by the geographic groupings above. For example, the UK Biobank, while predominantly of European ancestry, has representation of individuals with ancestry from each of the five regions used here (*Bycroft et al., 2018*). The 1KGP also sampled South Asian ancestry from multiple locations outside of South Asia, and whether those individuals show excess allele sharing due to recent admixture in those contexts is unclear. While we expect overall similar patterns to those seen here using emerging alternative datasets (*Bergström et al., 2019*), there may be subtle differences due to sampling and study design considerations.

Prior representations of human genetic variation data similar to the one presented here can be found in *Zietkiewicz et al., 1998*, who showed patterns of absence/presence/fixation at seven sites in the dys44 locus using a gray-scale, in a manner similar to *Figure 1* here. Other previous examples depict the proportion of variants with different geographic distributions resolved at the level of presence/absence (e.g. *Rosenberg et al., 2002*, Supp Figure 1 [pie chart]; *Szpiech et al., 2008*, Table 1, [circular bar]; *Rosenberg, 2011*, Table 2, Figure 4 [pie chart] for microsatellites; and *Jakobsson et al., 2008*, Figure 1A [Venn diagram] for SNPs, haplotypes and copy number variants). Publications on recent whole-genome sequence data from humans have several related and relevant figures for understanding the geographic distribution of variants (e.g. 1000 Genomes 2012, *Figure 2B*; *Auton et al., 2015*, *Figures 1A* and *3A*; *Bergström et al., 2019*, *Figure 3A* and Visual Abstract). The GeoVar plots provide a complementary view to these previous figures. Specifically, they provide more fine-grained representation than dichotomizations into private vs. shared variants and assessments of sharing based on presence versus absence. The GeoVar plots also complement plots of doubleton sharing or alternative normalized metrics that lose interpretability in terms of absolute allele frequency patterns and the numbers of variants with particular patterns.

The visualizations provided here help reinforce the conclusions of a long history of empirical studies in human genetics (*Lewontin, 1972*; *Ramachandran et al., 2005*; *Conrad et al., 2006*; *Li et al., 2008*; *Auton et al., 2015*; *Mallick et al., 2016*; *Bergström et al., 2019*). The results show how the human population has an abundance of localized rare variants and broadly shared common variants, with a paucity of private, locally common variants. Together these are footprints of the recent common ancestry of all human groups. As a consequence, human individuals most often differ from one another due to common variants that are found across the globe. Finally, although not examined explicitly above, the large abundance of rare variants observed here is another key feature of human variation and a consequence of recent human population growth (*Slatkin and Hudson, 1991*; *Di Rienzo and Wilson, 1991*; *Keinan and Clark, 2012*; *Nelson et al., 2012*; *Tennessen et al., 2012*).

The well-established introgression of archaic hominids (e.g. Neandertals, Denisovans) into modern human populations (*Wolf and Akey, 2018*) is not apparent in the GeoVar plots we produced. We believe that there are two broad reasons for this: (1) The clearest signal of archaic introgression will come from sites where archaic hominids differed from modern humans, and we expect that these sites are only a very small fraction of variants found in humans today. The average human–Neandertal and human–Denisovan sequence divergence are both less than 0.16% (using observations from *Prüfer et al., 2014*), and a recent study estimates that there are fewer than 70 Mb (2.3% of the genome) of Neanderthal introgressed segments per individual for all individuals in the 1KGP (*Chen et al., 2020*). (2) We do not expect SNVs from archaic introgression to be concentrated in a single GeoVar category. For example, introgressed variants occupy a wide range of allele frequencies (*Bergström et al., 2019*). Archaic introgression events are believed to be old: >30,000 years ago, allowing time for substantial genetic drift and admixture among human populations (*Chen et al., 2020*). Negative selection (*Harris and Nielsen, 2016*; *Juric et al., 2016*) and, in some cases, strong positive selection *Racimo et al., 2015* have also shaped the patterns of introgressed SNVs. For these reasons, we expect low levels of archaic introgression not to create a striking visual deviation in our GeoVar plots from the background patterns of a Recent African Origin model with subsequent migration (*Box 2*). To highlight the contributions of archaic hominids to human variation, more targeted approaches are needed (e.g. *Green et al., 2010*; *Durand et al., 2011*). Future work could also naturally extend the approach here to include archaic sequence data.

The geographic distributions of genetic variants visualized here are relevant for a number of applications, including studying geographically varying selection (*Yi et al., 2010*; *Key et al., 2018*), human demographic history (*Gutenkunst et al., 2009*), and the genetics of disease risk. For instance, due to ascertainment bias in arrays (*Figure 5*) and power considerations, common variants are often found in genome-wide association studies of disease traits (*Manolio et al., 2009*). The patterns shown above make it clear that most common variants are shared across geographic regions. Indeed, many common variant associations replicate across populations (*Marigorta and Navarro, 2013*; though see *Martin et al., 2017*; *Mostafavi et al., 2020* for complications). More recently, due to increasing sample sizes and sequencing-based approaches, disease mapping studies are finding more associations with rare variants (*Bomba et al., 2017*). As our work here emphasizes, rare variants are likely to be geographically restricted, and so one can expect the rare variants found in one population will not be useful for explaining trait variation in other populations, although they may identify relevant biological pathways that are shared across populations.

A future direction for the work here would be to apply our approach to other classes of genetic variants such as insertions, deletions, microsatellites, and structural variants. We note that in studies with sample sizes similar to or smaller than the 1KGP, nearly all SNVs arise from single mutation events. For other variants that arise from single mutation events (e.g. indels that arise from single mutations), we expect similar patterns to those observed for SNVs here. In contrast, for highly mutable loci we expect independently derived alleles will be distributed in disjoint regions of the world due to multiple mutational origins (*Ralph and Coop, 2010*).

Another future direction would be to shift from visualizing patterns of allele sharing to the patterns of sharing of ancestral lineages in coalescent genealogies. Recent advances in the inference of genome-wide tree sequences (*Kelleher et al., 2019*; *Speidel et al., 2019*) and allele ages (*Albers and McVean, 2020*) allow for quantitative summaries of ancestral lineage sharing. Such quantities have a close relationship to the multi-population SFS properties that are studied here, yet are more fundamental in a sense and less subject to the stochasticity of the mutation process. That said, the conceptual simplicity of visualizing allele frequency patterns may be an advantage in educational settings.

Most importantly, future applications of the approach to humans will ideally use datasets that include a greater sampling of the world's genetic diversity (*Bustamante et al., 2011*; *Popejoy and Fullerton, 2016*; *Martin et al., 2017*; *Peterson et al., 2019*). A related point is that the application of our method to genotyping array variants (*Figure 5*) reinforces the importance of considering the ancestry of study populations in genotype array design and selection (*Peterson et al., 2019*).

While we have focused here on human diversity at a global scale, GeoVar plots may be a useful tool for population geneticists working at other scales and with other species. The input to the visualization is simple: a table of allele frequencies in a set of populations. In the GeoVar software package, we provide python code for generating this table from a vcf file and a table of population

labels, but the user could generate the input from other data instead. For studying population structure, it is best to use an unbiased estimate of allele frequencies from, for example, whole-genome or reduced-representation sequencing.

Applied to new data sets, GeoVar may be used for exploratory data analysis, allowing users to see some important features of population structure without fitting explicit models. For example, hierarchical structure (*Figure 6*, rare variants shared within regional groupings) and recent admixture (*Figure 3*, rare variants shared between AFR and AMR) show up as distinctive patterns in the plots. *Box 2* shows that when the cutoff frequency separating Rare from Common mutations is close to the population split time (measured in units of 2$N$), an enrichment of 'RU' and 'CC' codes is expected. For example, in populations that split $0.1 \times N$ generations ago, mutations at local frequencies below 0.1 will tend to be private and those at higher frequencies will tend to be shared. In spatially distributed populations with limited dispersal, we expect that a similar relationship exists between cutoff frequencies, variant sharing patterns, and the geographic distance between populations. In an exploratory setting, users could generate plots with multiple cutoff frequencies to reveal varying levels of structure among populations. GeoVar plots may also serve as an informal goodness-of-fit check for parametric models of population history (as in *Figure 3—figure supplement 2*). In such exploratory and model-checking applications, attention to sample sizes and their configuration across sampling units is important, as larger sample sizes will allow the detection of more rare variants (e.g. contrast *Figure 3—figure supplement 2*, panel A and B). For the application to humans shown here, a preliminary approach to account for varying sample size did not substantially change the results (results not shown); that said, developing such an approach more fully or taking rarefaction approaches (*Szpiech et al., 2008*) may be essential for future applications with more uneven sample sizes.

Overall, the visualizations produced here provide an interpretable way to depict geographic patterns of human genetic variation. With personal genomic technologies and ancestry testing becoming commonplace, there is increasing importance in fostering the understanding of human population genetics. To this end, human genetics researchers must develop interpretable materials on patterns of genetic variation for use in educational and outreach settings (*Donovan et al., 2019*). The variant-centric approach detailed here complements existing visualizations of population structure, facilitating a clearer understanding of the major patterns of human genetic diversity.

## Acknowledgements

The 1KGP data downloaded and used here were generated by the New York Genome Center with funds provided by NHGRI Grant 3UM1HG008901-03S1. We thank members of the Novembre Lab, especially as this project was initiated in a group hackathon with contributions from Hussein Al-Asadi, Kushal Dey, Evan Koch, Joe Marcus, Ben Peter, Mark Reppell, and Joel Smith. We also thank Jeremy Berg, Jedidiah Carlson, Anna Di Rienzo, Joe Marcus, Aaron Panofsky, Molly Przeworski, Harald Ringbauer, Noah Rosenberg, Mashaal Sohail, Matthias Steinrücken, Paul Strode, Danny Townsend, and Xin He for comments on the manuscript draft, and Brian Donovan for additional helpful conversations. We thank Chi-Chun Liu and Vivaswat Shastry for comments on the GeoVar software package. This work was completed in part with resources provided by the University of Chicago's Research Computing Center and was supported by NIH training grant T32 GM07197 (AB), the University of Chicago 'Chicago Fellows' program (DPR), and NIH grant R01 GM132383.

## Additional information

### Funding

| Funder | Grant reference number | Author |
| --- | --- | --- |
| National Institute of General Medical Sciences | R01 GM132383 | Arjun Biddanda<br>John Novembre |
| Chicago Fellows Program of the University of Chicago | | Daniel P Rice |
| National Institute of General Medical Sciences | T32 GM07197 | Arjun Biddanda |

The funders had no role in study design, data collection and interpretation, or the decision to submit the work for publication.

### Author contributions
Arjun Biddanda, Conceptualization, Data curation, Software, Investigation, Visualization, Methodology, Writing - original draft, Writing - review and editing; Daniel P Rice, Conceptualization, Software, Formal analysis, Methodology, Writing - original draft, Writing - review and editing; John Novembre, Conceptualization, Supervision, Funding acquisition, Visualization, Writing - original draft, Project administration, Writing - review and editing

### Author ORCIDs
Arjun Biddanda (ID) https://orcid.org/0000-0003-1861-1523
Daniel P Rice (ID) https://orcid.org/0000-0002-9509-2694
John Novembre (ID) https://orcid.org/0000-0001-5345-0214

### Ethics
Human subjects: This work analyzes anonymized publicly available data consented for studies of population genetic variation.

### Decision letter and Author response
Decision letter https://doi.org/10.7554/eLife.60107.sa1
Author response https://doi.org/10.7554/eLife.60107.sa2

## Additional files
### Supplementary files
• Supplementary file 1. Table S1. Population abbreviations and groupings used in the study.

• Transparent reporting form

### Data availability
The GeoVar assignments for each variant have been deposited to Dryad (https://doi.org/10.5061/dryad.rjdfn2z7v). The code for replicating the analyses is available at: https://github.com/aabiddanda/geovar_rep_paper (copy archived at https://archive.softwareheritage.org/swh:1:rev:db3ca8-faeecf8697973f803bc05c5a3d0a187145/). A python package (https://aabiddanda.github.io/geovar/) allows users to make GeoVar plots from frequency tables or VCF files.

The following dataset was generated:

| Author(s) | Year | Dataset title | Dataset URL | Database and Identifier |
|---|---|---|---|---|
| Biddanda A, Rice DP, Novembre J | 2020 | Geographic allele frequency variation in the 1000 Genomes hg38 NYGC dataset | https://doi.org/10.5061/dryad.rjdfn2z7v | Dryad Digital Repository, 10.5061/dryad.rjdfn2z7v |

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

## Appendix 1

### Theoretical geographic distribution code abundances

The relative abundances of geographic distribution codes derive from human population history (**Box 2**). Here, we use a simple population genetic model to develop intuition about the relationship between the divergence time of a pair of populations and the expected two-letter code abundances. To isolate the effect of population divergence from other factors such as population growth, we consider the simplest possible model of divergence: two constant-size populations of $N$ individuals descended from a single $N$-individual source population $T$ generations ago (**Box 2—figure 1A**). We incorporate recent contact between populations via a symmetric admixture coefficient $\alpha$. Individuals in Population 1 derive a fraction $\alpha$ of their ancestry from Population 2 and vice versa. Human population history is much more complex than our model, but it captures the essential features of common ancestry, subsequent isolation, and modern admixture.

Python source code implementing the calculation and producing **Box 2—figure 1** is available in the project's Git repository (https://github.com/aabiddanda/geovar_rep_paper; **Biddanda, 2020b**; copy archived at swh:1:rev:db3ca8faeecf8697973f803bc05c5a3d0a187145).

## Wright-Fisher diffusion of allele frequencies

In our model, allele frequencies in the two source populations are initially identical because they derive from the same source population. After the populations split, allele frequencies evolve independently according to a Wright-Fisher diffusion with symmetric mutations at rate $\theta$ new mutations per population per generation. At time $t = T/2N$ generations after the split, the joint density of mutations at frequency $x_1$ in Population 1 and $x_2$ in Population 2 is given by,

$$f(t; x_1, x_2) = \int_0^1 f(0; x_0) p(t; x_0, x_1) p(t; x_0, x_2) dx_0, \tag{1}$$

where $f(0; x_0)$ is the density of mutations at frequency $x_0$ in the source population and $p(t; \cdot, \cdot)$ is the Wright-Fisher transition density function. Assuming that the source population was at mutation-drift equilibrium, $f(0; x_0) = \pi(x_0) \propto (x_0(1 - x_0))^{\theta - 1}$, the stationary measure of the Wright-Fisher diffusion.

We use the spectral decomposition of **Song and Steinrücken, 2012** to represent the Wright-Fisher transition density as an infinite sum of modified Jacobi polynomials, $B_i(x)$:

$$p(t; x, y) = \sum_{i=0}^{\infty} e^{-\Lambda_i t} \pi(y) \frac{B_i(x) B_i(y)}{\langle B_i, B_i \rangle}, \tag{2}$$

where the inner product $\langle g, h \rangle$ is given by $\int_0^1 f(x) g(x) \pi(x) dx$. The Jacobi polynomials are orthogonal with respect to this inner product. That is, $\langle B_i, B_j \rangle = 0$ for $i \neq j$. Substituting (2) into (1) and using orthogonality, we have:

$$f(t; x_1, x_2) = \pi(x_1) \pi(x_2) \sum_{i=0}^{\infty} e^{-2\Lambda_i t} \frac{B_i(x_1) B_i(x_2)}{\langle B_i, B_i \rangle}. \tag{3}$$

In practice, we can only compute partial sums on the right-hand side, which we can re-write as

$$f(t; x_1, x_2) = \pi(x_1) \pi(x_2)(S_m(x_1, x_2) + R_m(x_1, x_2)), \tag{4}$$

where $S_m$ is the partial sum of terms up to order $m$ and $R_m$ is the remainder, which represents the error from truncating the series. We can control this error by choosing a large enough $m$ (see Numerical Integration.)

## Sampling probabilities

The abundances of two-population distribution codes is a simple transformation of the cumulative distribution function (CDF) of the joint allele counts $(K_1, K_2)$. Conditioning on allele frequencies at time $t$, but before admixture, the CDF is given by

$$\mathcal{P}\{K_1 \le k_1, K_2 \le k_2\} = \int_0^1 \int_0^1 \mathcal{P}\{K_1 \le k_1 | x_1, x_2\} \mathcal{P}\{K_2 \le k_2 | x_1, x_2\} f(t; x_1, x_2) dx_1 dx_2 \qquad (5)$$

For $n$ randomly sampled haploid individuals from each population, and admixture coefficient $\alpha$, we have:

$$K_1 | x_1, x_2 \sim \mathrm{Binomial}(n, (1-\alpha)x_1 + \alpha x_2),$$

$$K_2 | x_1, x_2 \sim \mathrm{Binomial}(n, (1-\alpha)x_2 + \alpha x_1).$$

Writing $P_n^{(k)}(x_1, x_2)$ for the binomial cumulative distribution function $\mathcal{P}\{K_i \le k | x_1, x_2\}$, and substituting (5) into (4) yields:

$$\mathcal{P}\{K_1 \le k_1, K_2 \le k_2\} = \left\langle P_n^{(k_1)} P_n^{(k_2)}, S_m \right\rangle + \left\langle P_n^{(k_1)} P_n^{(k_2)}, R_m \right\rangle \qquad (6)$$

where the inner product now represents the double integral weighted by $\pi(x_1)\pi(x_2)$.

## Numerical integration

We compute the integrals in (6) by two-dimensional Gauss-Jacobi quadrature. The left argument of the inner product is a polynomial of degree $n$ in both $x_1$ and $x_2$. As a result, we can choose $m = 2n$, so that $\left\langle P_n^{(k_1)} P_n^{(k_2)}, R_{2n} \right\rangle = 0$ due to the orthogonality of the Jacobi polynomials. Because $S_{2n}$ is also a polynomial, the integrand is a polynomial of degree $4n$. Thus, fixed-order tensor-product Gauss-Jacobi quadrature is guaranteed to yield the exact integral with $4n^2$ evaluations of the integrand.

## Appendix 2

### Extinction probability and conditional mean frequency

The extinction probability $\wp(p,t)$, the probability that a mutation that was at frequency $p$ at time $t = 0$ is extinct at time $t = T/2N$, obeys the Kolmogorov backward equation *Ewens, 2004* :

$$\frac{\partial}{\partial t}\wp(p,t) = \frac{1}{2}p(1-p)\frac{\partial^2}{\partial p^2}\wp(p,t) \tag{7}$$

with boundary conditions

$$\wp(p,0) = \begin{cases} 1 & \text{if } p\text{=}0 \\ 0 & \text{otherwise} \end{cases} \tag{8}$$

$$\wp(0,t) = 1 \tag{9}$$

$$\wp(1,t) = 0 \tag{10}$$

For short times and rare alleles (i.e. $t, p \ll 1$), we can use the approximation $p(1-p) \approx p$, to get a simpler diffusion equation:

$$\frac{\partial}{\partial t}\wp = \frac{1}{2}p\frac{\partial^2}{\partial p^2}\wp \tag{11}$$

with modified boundary conditions

$$\wp(p,0) = \begin{cases} 1 & \text{if } p\text{=}0 \\ 0 & \text{otherwise} \end{cases} \tag{12}$$

$$\wp(0,t) = 1 \tag{13}$$

$$\lim_{p \to \infty} \wp(p,t) = 0 \tag{14}$$

Because we are neglecting the $(1-p)$ term, fixation is not possible in this approximation, and it is natural to move the upper boundary condition from $p = 1$ to $p \to \infty$. (This approximation is equivalent to replacing the Wright-Fisher diffusion with a continuous-state critical branching process, which is guaranteed to go extinct for all finite sizes). Accordingly, we expect the approximation to break down when the minor allele has a substantial probability of fixation.

We can solve (11) in closed form to find the time-dependent extinction probability,

$$\wp(p,t) \approx \exp\left(-\frac{2p}{t}\right), \tag{15}$$

For $t \ll 2p$, this probability is exponentially small, while for $t > 2p$ it behaves like $1 - 2p/t$ (*Box 2— figure 1C*).

We can use (15) to find the expected frequency of a new mutation conditional on its survival to time *t*. By the law of total probability, we have

$$\mathbb{E}[X(t)|X(t)\text{>}0] = \frac{\mathbb{E}[X(t)]}{\mathbb{P}[X(t)\text{>}0]} = \frac{1/2N}{1 - \wp(1/2N, t)}, \tag{16}$$

where in the last equality we used the fact that for a new neutral mutation $\mathbb{E}[X(t)] = p = 1/2N$. Thus, to leading order in $1/N$, we have $\mathbb{E}[X(t)|X(t)\text{>}0] \sim t/2$.

# Appendix 3

**Appendix 3—key resources table**

| Reagent type (species) or resource | Designation | Source or reference | Identifiers | Additional information |
|---|---|---|---|---|
| Other | 1000 Genomes High-Coverage Data (1 KG) | https://doi.org/10.1093/nar/gkz836 | RRID:SCR_006828 | http://ftp.1000genomes.ebi.ac.uk/vol1/ftp/data_collections/1000G_2504_high_coverage/working/20190425_NYGC_GATK/ |
| Other | Simons Genome Diversity Project Data (SGDP) | https://doi.org/10.1038/nature18964 | | https://reichdata.hms.harvard.edu/pub/datasets/sgdp/ |
| Other | Ancestral allele calls | https://doi.org/10.1093/nar/gkz966 | RRID:SCR_002344 | ftp.ensembl.org/pub/release-90/fasta/ancestral_alleles/homo_sapiens_ ancestor_GRCh38_e86.tar.gz |
| Other | GrCH38 Genome Masks | https://doi.org/10.1093/nar/gkz836 | RRID:SCR_006828 | http://ftp.1000genomes.ebi.ac.uk/vol1/ftp/data_collections/1000_genomes_project/working/20160622_genome_mask_GRCh38/ |
| Commercial assay or kit | Human Origins Array; Human Origins | other | | https://sec-assets.thermofisher.com/TFS-Assets/LSG/Support-Files/Axiom_GW_%20HuOrigin.na35.annot.csv.zip |
| Commercial assay or kit | Affymetrix GenomeWide 6.0 Array (Affy6) | other | | http://www.affymetrix.com/Auth/analysis/downloads/na35/genotyping/GenomeWideSNP_6.na35.annot.csv.zip |
| Commercial assay or kit | Illumina MEGA Array (MEGA) | other | | https://support.illumina.com/array/array_kits/infinium-multi-ethnic-amr-afr-8-kit/downloads.html |
| Commercial assay or kit | Illumina Human Omni Express Array (OmniExpress) | other | | ftp://ussd-ftp.illumina.com/Downloads/ProductFiles/HumanOmniExpress-24/v1-0/HumanOmniExpress-24-v1-0-B.csv |
| Commercial assay or kit | Illumina Omni2.5Exome Array (Omni2.5Exome) | other | | ftp://ussd-ftp.illumina.com/Downloads/ProductFiles/HumanOmni2-5Exome-8/Product_Files_v1-1/HumanOmni2-5Exome-8-v1-1-A.csv |
| Other | Reproducible analysis pipeline for this paper | This paper | | https://github.com/aabiddanda/geovar_rep_paper; *Biddanda, 2020a* (copy archived at swh:1:rev:db3ca8faeecf8697973f803bc05c5a3d0a187145) |
| Software, algorithm | GeoVar software | This paper | | https://aabiddanda.github.io/geovar/ |

