## [Decision Letter]

**Acceptance summary:**

This paper reframes traditional summaries of human genetic variation into an interpretable visual framework. This visualization emphasizes how variation is shared across geography, and links these patterns to the evolutionary processes generating variation. Importantly, the authors discuss the implications for public and scientific misconceptions about human evolution and genetic differentiation.

**Decision letter after peer review:**

Thank you for submitting your article "Geographic patterns of human allele frequency variation: a variant-centric perspective" for consideration by *eLife*. Your article has been reviewed by three peer reviewers, one of whom is a member of our Board of Reviewing Editors, and the evaluation has been overseen by George Perry as the Senior Editor. The reviewers have opted to remain anonymous.

The reviewers have discussed the reviews with one another and the Reviewing Editor has drafted this decision to help you prepare a revised submission.

Summary:

This manuscript reports an interesting perspective on the distribution of genetic variation in human populations through a new visualization. Focusing on the geographic distribution of single variants, the authors provide a framework that easily summarizes global patterns of variation in a way that is intuitive and perhaps less prone to certain visual biases. This is particularly important to broaden discussions of human genetic variation to non-specialists. The approach largely captures the information of joint SFS, allowing more populations. The context and discussion of implications is a real strength of this paper, which bridges a tool and a synthesis paper. With a clearer workflow from data to visualization and improved documentation, it will be a useful visualization tool. Overall it is well-written and easy to read, with care taken to consider details such as the role of singletons and data type.

Essential revisions:

1) Applicability and availability as a tool/resource. Would this approach be useful for non-humans? What types of relevant questions could you answer, and what datasets would be necessary? Similarly, how would this type of visualization be interpreted or used for, e.g. within-continent studies in humans? Sample size and sampling scheme is only briefly mentioned, though may be particularly impactful when attempting to apply this to other contexts.

Importantly, the github repository should be set up in a way that would facilitate reuse in other datasets. It would be nice to see a tutorial running through using the software on a small example dataset on the github repo or supplement, or other attempts to make this more user-friendly.

2) Connections to previous work. The idea is reminiscent of historical population genetic studies which often focused on single or only a few variants because of sequencing limitations. It may be useful to include more historical citations or perspectives. Similarly, there are other attempts to visualize this kind of information, for example Figure 1B of the 1000 Genomes 2012 paper, Figure 1A of the 2015 paper, Figure 4A of Bergstrom et al., 2020, and corresponding statements in those papers. These broadly show the same things as the present paper. It would help to be better to be more explicit about what this visualization adds in the context of previous work.

3) Models of population history. A key point of the paper is connecting the visualized patterns to models of human history, yet the models considered are fairly simplistic. Specifically, in Figure 4, the models do not include features like bottlenecks, recent exponential growth, and extensive gene flow, all of which would be expected to change the patterns observed. It would help to either include simulations or discussion on how these common features may interact with the observations presented in the paper.

Notably, it is a mischaracterization to equate the first model to the multiregional model (subsection “Box 2: Theoretical Modeling”). Wolpoff's multiregional model emphasized (1) extensive gene flow and (2) parallel adaptation, recognizing that these were necessary to explain the low Fst and phenotypic differentiation among regions. See, for example, Wolpoff et al. "Multiregional Evolution: A World-Wide Source for Modern Human Populations" in Origins of Anatomically Modern Humans" eds. Nitecki and Nitecki, Springer 1994. Since the model doesn't include these it's really rejecting something like Coon's models from the 1950s, which even 30 years ago were widely recognized to be implausible. That said, simulating a more Wolpoff-like model seems unnecessary, since that debate is largely settled. Text (and perhaps figure) clarifications can improve accuracy and discussion for non-science audiences along these lines may better fit in the Discussion.

---

## [Author Response]

Essential revisions:1) Applicability and availability as a tool/resource. Would this approach be useful for non-humans? What types of relevant questions could you answer, and what datasets would be necessary? Similarly, how would this type of visualization be interpreted or used for, e.g. within-continent studies in humans? Sample size and sampling scheme is only briefly mentioned, though may be particularly impactful when attempting to apply this to other contexts.

Great questions. We have added the following paragraphs to the Discussion. We have also run simulations for scenarios that differ only in the sample sizes used (see Figure 3—figure supplement 4 and response to 3b).

“While we have focused here on human diversity at a global scale, GeoVar plots may be a useful tool for population geneticists working at other scales and with other species. […] In such exploratory and model-checking applications, attention to sample sizes and their configuration across sampling units is important, as larger sample sizes will allow the detection of more rare variants (e.g., contrast Figure 3—figure supplement 4A and B).”

Importantly, the github repository should be set up in a way that would facilitate reuse in other datasets. It would be nice to see a tutorial running through using the software on a small example dataset on the github repo or supplement, or other attempts to make this more user-friendly.

We have published a python package (https://aabiddanda.github.io/geovar/) that allows users to make GeoVar plots from frequency tables or VCF files. The documentation includes a jupyter notebook tutorial, which can be run remotely from the project’s GitHub page on the notebook hosting service mybinder.org. The package includes an example data set: 5000 biallelic 1KGP variants from chromosome 22. We have added this repository URL to our revised manuscript.

2) Connections to previous work. The idea is reminiscent of historical population genetic studies which often focused on single or only a few variants because of sequencing limitations. It may be useful to include more historical citations or perspectives. Similarly, there are other attempts to visualize this kind of information, for example Figure 1B of the 1000 Genomes 2012 paper, Figure 1A of the 2015 paper, Figure 4A of Bergstrom et al., 2020, and corresponding statements in those papers. These broadly show the same things as the present paper. It would help to be better to be more explicit about what this visualization adds in the context of previous work.

This is an important point, and we now elaborate connections to previous work more elaborately in the paper with the following remarks in a new paragraph of the discussion:

“Prior representations of human genetic variation data similar to the one presented here can be found in Zietkiewicz et al., 1998, who showed patterns of absence/presence/fixation at seven sites in the dys44 locus using a gray-scale, in a manner similar to Figure 1 here. […] The GeoVar plots also complement plots of doubleton sharing or alternative normalized metrics that lose interpretability in terms of absolute allele frequency patterns and the numbers of variants with particular patterns.”

3) Models of population history. A key point of the paper is connecting the visualized patterns to models of human history, yet the models considered are fairly simplistic. Specifically, in Figure 4, the models do not include features like bottlenecks, recent exponential growth, and extensive gene flow, all of which would be expected to change the patterns observed. It would help to either include simulations or discussion on how these common features may interact with the observations presented in the paper.

The reviewer is correct that the observed allele frequency patterns depend on the full population history, rather than only on the split time. To illustrate this, we have simulated the models of Gutenkunst et al., 2009, and Tennessen et al., 2012, via the stdpopsim library (Adrion et al., 2020) and computed the expected GeoVar patterns (Box 2 – Figure 1—figure supplement 1). The simulations recapitulate the main point of Box 2: that the prevalence of RU and CC codes is predicted by models with shallow splits. They also show that the exact proportions of codes depend on the model and sampling scheme, and we emphasize this point further in the Discussion (see response to 1a).

Notably, it is a mischaracterization to equate the first model to the multiregional model (subsection “Box 2: Theoretical Modeling”). Wolpoff's multiregional model emphasized (1) extensive gene flow and (2) parallel adaptation, recognizing that these were necessary to explain the low Fst and phenotypic differentiation among regions. See, for example, Wolpoff et al. "Multiregional Evolution: A World-Wide Source for Modern Human Populations" in Origins of Anatomically Modern Humans" eds. Nitecki and Nitecki, Springer 1994. Since the model doesn't include these it's really rejecting something like Coon's models from the 1950s, which even 30 years ago were widely recognized to be implausible. That said, simulating a more Wolpoff-like model seems unnecessary, since that debate is largely settled. Text (and perhaps figure) clarifications can improve accuracy and discussion for non-science audiences along these lines may better fit in the Discussion.

We apologize –this reflected our confusion inherited from an old review that described a version of the multi-regional hypothesis with deep divergence and no migration, akin to what we use in Box 1. We appreciate the correction of our mistake in citing Wolpoff for that model and have changed the citation to Coon 1962: “The deep case mimics inaccurate models of human evolution with very ancient population splits (e.g., Coon 1962).”